# The periaqueductal gray and Bayesian integration in placebo analgesia

**Arvina Grahl\*, Selim Onat, Christian Büchel**

Department of Systems Neuroscience, University Medical Center Hamburg-Eppendorf, Hamburg, Germany

**Abstract** In placebo hypoalgesia research, the strength of treatment expectations and experiences are key components. However, the reliability or precision of expectations had been mostly ignored although being a likely source for interindividual differences. In the present study, we adopted a Bayesian framework, naturally combining expectation magnitudes and precisions. This postulates that expectations (prior) are integrated with incoming nociceptive information (likelihood) and both are weighted by their relative precision to form the pain percept and placebo effect. Sixty-two healthy subjects received heat pain during fMRI. Placebo effects were more pronounced in subjects with more precise treatment expectations and correlated positively with the relative precision of the prior expectation. Neural correlates of this precision were observed in the periaqueductal gray and the rostral ventromedial medulla, indicating that already at the level of the brainstem the precision of an expectation can influence pain perception presenting strong evidence for Bayesian integration in placebo hypoalgesia.

DOI: https://doi.org/10.7554/eLife.32930.001

## Introduction

Placebo effects and concomitant hypoalgesia in the pain context exemplify the substantial influence that expectation and experience can have on treatment outcomes and have therefore been intensely investigated over the last decades (*Atlas et al., 2010*; *Colloca and Benedetti, 2006*; *de la Fuente-Fernández et al., 2001*; *Enck et al., 2013*; *Kirsch, 1999*; *Reicherts et al., 2016*; *Rief et al., 2011*; *Schenk et al., 2014*; *Stone et al., 2005*; *Wager et al., 2004*). Various studies have identified factors that modulate these effects such as value (*Geuter et al., 2013*; *Waber et al., 2008*), treatment history (*Kessner et al., 2014*; *Kessner et al., 2013*), doctor-patient relationship (*Benedetti, 2013*), and context effects (*Blasi et al., 2001*). Importantly, these modulators are likely to affect the precision or inverse variance of those expectations, which might explain – at least in part – the large interindividual differences observed in placebo hypoalgesia studies (*Vase et al., 2009*; *Wager et al., 2011*). For that reason, it is presumed that different precision levels in expectations and prior treatment experience seem to potentially change treatment outcomes (*Büchel et al., 2014*). Pollo and colleagues presented different verbal instructions to manipulate patients' treatment response expectancies and observed the largest analgesic effect in the 'sure treatment' group compared to two other groups being either non-informed or having only a 50% chance of receiving a painkiller (*Pollo et al., 2001*). This manipulation of the probability of treatment indicates that this is likely to have an effect on the ensuing placebo response. The more precise expectations are the more probable a specific treatment outcome might be presumed. More recent approaches or frameworks in pain research already focused on accounting for several modulating factors influencing pain perception to provide a broader view on this very subjective topic (*Anchisi and Zanon, 2015*; *Büchel et al., 2014*; *Petzschner et al., 2017*; *Wager et al., 2013*; *Wiech et al., 2014*).

For that reason, this was a starting point to formally develop a model which adequately combines the precision and magnitude of prior expectations on placebo effects (*Büchel et al., 2014*) and test

**\*For correspondence:**
a.grahl@uke.de

**Competing interests:** The authors declare that no competing interests exist.

**eLife digest** On a battlefield in World War II, surgeon Henry Beecher ran out of morphine. To his surprise, he found that replacing the missing morphine with saltwater allowed him to continue operating on wounded soldiers. Although saltwater contains no active pain-relieving ingredients, it reduced the soldiers' pain. This is an example of the placebo effect. Placebos have been shown to reduce autonomic responses to pain, such as sweating. They also modulate activity in brain regions that process pain. But why do some of us experience larger placebo effects than others?

Grahl et al. propose that the size of the placebo effect depends on our expectations about a treatment. More specifically, it depends on how precise those expectations are. Imagine two people who have taken the same treatment many times, and who have experienced the same average reduction in pain. But for one person, the treatment reduced their pain by roughly the same amount each time. For the other, the treatment sometimes reduced their pain by a large amount and other times hardly at all. The first person will have more precise expectations than the second about how effective the treatment will be in future. Grahl et al. propose that the first person will thus experience a greater placebo effect in response to a 'fake' version of the treatment.

To test this idea, Grahl et al. applied painful heat to the forearms of healthy volunteers lying inside a brain scanner. On half the trials, the volunteers were told that they would also receive an electrical pain-relieving therapy. In reality, this treatment was never applied. After each trial, the volunteers rated the intensity of the pain they had experienced. As expected, the volunteers reported less pain when they thought they were receiving a pain-relieving treatment. Moreover, those volunteers with more precise expectations about the treatment reported greater pain relief than volunteers with less precise expectations. The former group also showed less activity in one of the brain's major pain-processing centers, the periaqueductal gray.

These findings help shed light on why some people experience larger placebo effects than others. They suggest that helping patients form precise expectations about their treatment, by giving them precise information about its likely effectiveness, may boost the placebo effect. Further studies are needed to determine whether this phenomenon also occurs in patients with pain disorders. If it does, it could help such patients manage their pain using fewer active painkillers.
DOI: https://doi.org/10.7554/eLife.32930.002

this in a group of individuals. At the heart of this model is optimal Bayesian integration (*Knill and Pouget, 2004*; *Körding and Wolpert, 2004*; *O'Reilly et al., 2012*), a framework that explicitly accounts for variability of prior information such as expectations and might thus be helpful to explain the integration of expectations and sensory information in placebo hypoalgesia (*Büchel et al., 2014*). Implicit in this idea is the assumption, that the brain constantly integrates incoming sensory input with expectations and, as a consequence, generates new expectations about the environment to minimize future surprise (*Feldman and Friston, 2010*; *Friston, 2010*; *Friston and Kiebel, 2009*). In essence, Bayesian integration optimally integrates previous expectations (prior) with incoming sensory information (likelihood) and makes a prediction about the outcome of a certain event (posterior): the posterior is proportional to the product of prior and likelihood. Importantly, in this framework, both terms are weighted by their relative precision to estimate the posterior. Putting this into the placebo context, previous treatment experience and expectations serve as the Bayesian prior being illustrated by a probability distribution reflecting pain relief as well as treatment efficacy precision. A new incoming untreated pain experience reflects the likelihood distribution. By integrating the two Bayesian key components, prior and likelihood, this framework (*Büchel et al., 2014*) offers the opportunity to explain the outcome of a new treatment experience, including the placebo effect, by predicting ones perceived pain as the model posterior (*Figure 1A,B*):

$$p(pain|sensory\,input) \propto p(pain)*p(sensory\,input|pain) \tag{1}$$

A highly precise prior treatment expectation should lead to a stronger placebo effect as compared to a highly variable expectation. By collecting subjective pain perception in the form of explicit ratings (visual analogue scale, VAS), both, the mean and precision level (reflected by the

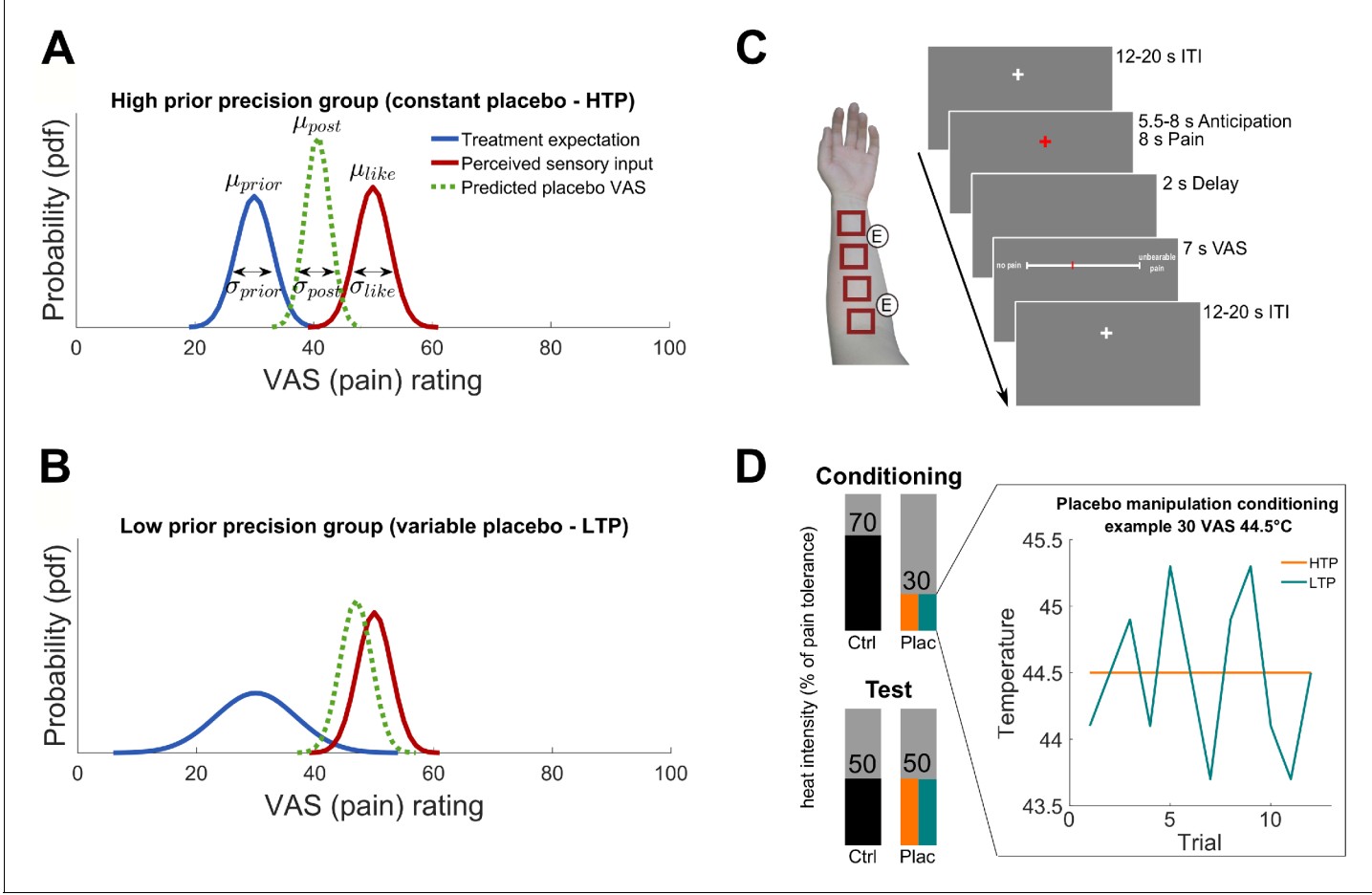

**Figure 1.** Hypotheses and experimental design. (A) and (B) Hypothesized posterior prediction per group illustrated by Gaussian probability density functions. The different prior distributions (blue) reflect precise (A) and variable (B) treatment expectations both around the conditioned VAS mean of 30 ($\mu_{prior}$) but with different standard deviations ($\sigma_{prior}$) due to the experimental manipulation. The likelihood distribution (red) was not manipulated and is therefore displayed identical for both groups. The posterior distribution (dashed green) reflects the model prediction of the perceived pain of a new treatment experience. In Bayesian integration, prior and likelihood are weighted by their relative precision which draws the respective prediction into the direction of the more precise distribution. The hypothesized placebo effect ($\mu_{like} - \mu_{post}$) is therefore larger for higher (A) compared to lower (B) treatment expectation precision. (C) Stimulation patches on the left arm including electrode positions for the sham TENS treatment and one experimental trial. (D) Conditioning and test phase pain intensities including the experimental prior manipulation of the two groups (see also *Figure 1—source data 1* and *Figure 1—figure supplement 1* for exit questionnaire information). An example is displayed representing a calibrated intensity of 30% pain tolerance at a temperature of 44.5°C. No variability was induced for the precise treatment expectation group (orange) compared to induced variability of SD 0.55°C for each subject of the variable treatment expectation group (turquoise). Mean temperature was the same for all participants irrespective of the group assignment. The heat pain intensity of conditioning control, test placebo as well as test control were always presented without induced variability in both groups. HTP, high treatment precision; LTP, low treatment precision; pdf, probability density function; μ, mean; σ, standard deviation; VAS, visual analogue scale; ITI, inter-trial-interval; ctrl, control condition; plac, placebo condition.
DOI: https://doi.org/10.7554/eLife.32930.003

The following source data and figure supplement are available for figure 1:

**Source data 1.** Post-experimental TENS-exit-questionnaire (prior to debriefing).
DOI: https://doi.org/10.7554/eLife.32930.005

**Figure supplement 1.** Frequencies of post-experimental TENS-questionnaire.
DOI: https://doi.org/10.7554/eLife.32930.004

variance) of these subjective reports translate into predictable changes in the pain percepts. These pain ratings are the basis of the aforementioned probability distributions.

Furthermore, numerous imaging studies identified brain regions being involved in processing placebo hypoalgesia (for a review see *Wager and Atlas, 2015*) highlighting the need for a multidimensional approach to investigate the underlying mechanisms. Models that can parsimoniously account

for several modulating factors (*Anchisi and Zanon, 2015*; *Büchel et al., 2014*; *Wager et al., 2013*; *Wiech et al., 2014*) are essential to better understand and relate behavioral as well as neuronal aspects of sensory processing such as pain. This can be achieved through model-based functional magnetic resonance imaging, fMRI (*Gläscher and O'Doherty, 2010*) by using various features of these models to identify related neural correlates.

In the present study, we directly tested the hypothesis that Bayesian optimal integration is a possible mechanism by which expectation is integrated with sensory information in the context of placebo hypoalgesia. To test this framework, we investigated the naturally occurring variability in expectations and also explicitly varied levels of expectation precision in two experimental groups (*Figure 1A,B*). This manipulation especially addressed the fact, that a pain treatment experience is usually not (only) verbally induced or influenced by a physician but mostly experienced by the perceived pain relief of the individual compared to the untreated nociceptive sensation. Therefore, treatment instructions had been identical for both groups but the treated painful sensation was manipulated in terms of the precision levels of prior treatment expectation. Signaled by two visual cues, participants of both groups received heat pain in a placebo treatment and untreated control condition (*Figure 1C,D*). Via different temperature variations across trials during a conditioning phase (*Figure 1D*), one group experienced the placebo treatment as variable (low treatment precision), whereas the other group experienced it as constant (high treatment precision). Assuming these two different priors represent different precision levels concerning expected treatment outcomes, we are able to compare them in terms of behavioral influences on observed placebo treatment outcomes. Importantly, we used fMRI to also investigate neural correlates of this mechanism in the brain adopting a model-based fMRI approach (*Gläscher and O'Doherty, 2010*). With this combined approach, we not only sought to better explain placebo treatment outcomes on a behavioral level, but also identify neural correlates which contribute to Bayesian integration of treatment expectations.

We focused our fMRI analysis primarily on the periaqueductal gray (PAG) and the underlying processing concerning the influence of precision levels of prior expectations on placebo treatment outcomes. This a priori region of interest (ROI) hypothesis was especially based on a previous study that implicated the PAG in processing of the precision of vicarious information (*Yoshida et al., 2013*). Supporting this, another study investigated pain avoidance prediction error coding (*Roy et al., 2014*) and showed that the PAG, among others, was involved in the modulation of expected probability of pain in this context. Combining this finding with the assumption that prediction error and precision level coding are distinct but share related aspects of modulatory functions, this is also hinting to the PAG being involved in the processing of prior treatment precision levels in a hypoalgesia context. Although the literature discusses several other regions such as lateral orbitofrontal (OFC) and rostral anterior cingulate cortex (rACC) to be involved in aversive prediction error coding (*Seymour et al., 2004*, *2005*; *Shackman et al., 2011*; *Zeidan et al., 2015*), yet, no other regions were specifically related to precision level or variability processing. For that reason, this study focused the imaging analysis on the PAG.

## Results

### Behavioral results

In a first step, we compared VAS ratings between the two groups during the conditioning and test phase respectively using a mixed-effects analysis. In this analysis, it was focused on the experimentally induced precision level of prior treatment expectation resulting in the use of a subset of participants ($N_{sub}$ = 49) ensuring that the intended conditioning manipulation of high vs. low prior treatment precision was induced according to the respective group assignment of subjects (high treatment precision - HTP: $n_{sub-HTP}$ = 23, low treatment precision - LTP: $n_{sub-LTP}$ = 26, see Material and methods for further description and generating procedure of sub-samples). As expected, for the conditioning phase (*Figure 2A*), a main effect of condition (placebo vs. control) was observed ($F_{(2,1172)}$ = 24.55, p<0.001) reflecting the two distinct temperatures used for creating the treatment experience. Neither a main effect of group (p=0.963) nor an interaction effect of group and condition (p=0.885) was revealed indicating that the conditioning procedure did not differ between the two groups. Subject-specific standard deviations for placebo conditioning ratings (*Figure 2B*) and

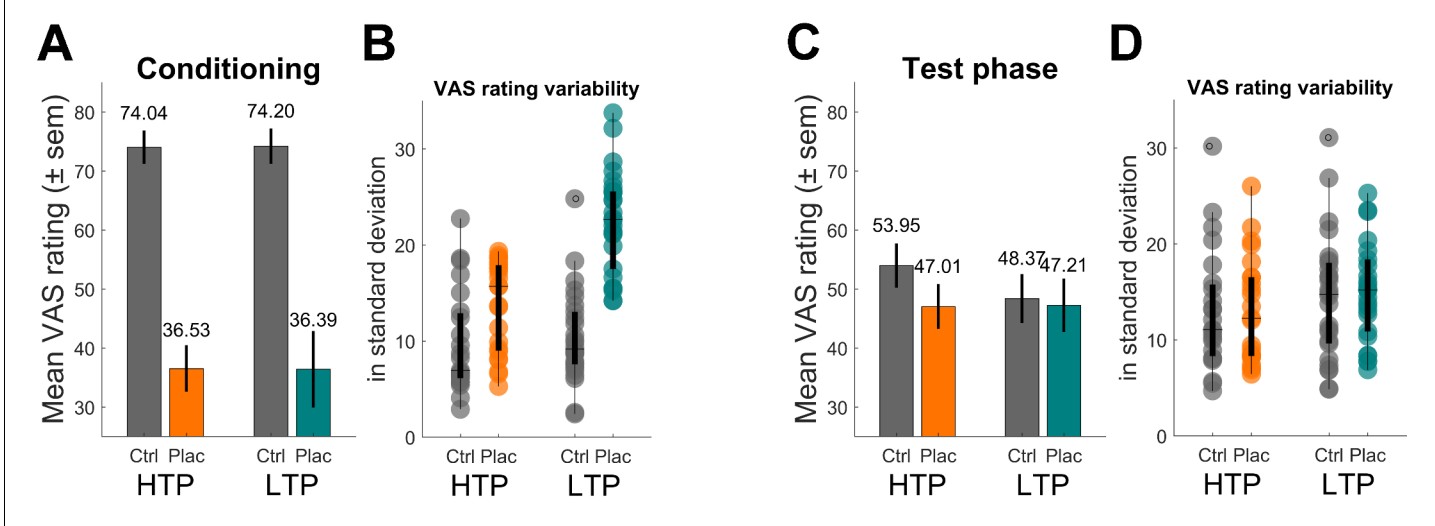

**Figure 2.** Behavioral results of pain ratings. (A) Conditioning phase pain ratings reflecting the two distinct temperatures for the control (ctrl) and placebo (plac) condition. (B) Boxplot of single subject standard deviations of pain ratings of the conditioning phase. Note the higher variability within the LTP group's placebo condition (turquoise). (C) Test phase pain ratings reflecting the analgesic effect of the treatment. The significant interaction effect revealed a larger placebo effect for the HTP compared to the LTP group. Note that the temperature for all trials of the two conditions was identical and constant for both groups. (D) Boxplot of single subject standard deviations of pain ratings of the test phase. Variability levels across conditions do not differ (D). HTP, high treatment precision ($n_{sub\text{-}HTP}$ = 23); LTP, low treatment precision ($n_{sub\text{-}LTP}$ = 26); sem, standard error of mean.
DOI: https://doi.org/10.7554/eLife.32930.006

therefore the variability (i.e. inverse precision) of prior treatment expectations was larger for the LTP compared to the HTP group (LTP 22.46 ± 5.33 vs. HTP 13.63 ± 4.76, t(47) = 6.12, p<0.001) reflecting that in their levels of placebo treatment precision the two groups differed. It is important to note that this effect of group differences concerning precision levels was driven by the use of a sub-sample in this analysis. This served as a proof of concept for the conditioning manipulation and was done to then investigate the precision level effect on the test phase placebo response (in other words, precision level being the independent compared to placebo effect being the dependent variable). Concerning the test phase (*Figure 2C,D*), in which all subjects received identical heat stimuli for both conditions, again, a main effect of condition was observed (F(2,1172) = 4.49, p<0.001). This reflects the significant placebo effect by indicating less painfulness for treatment compared to control stimuli even though participants received identical heat stimulation in both conditions. Importantly, the interaction between group and condition in the test phase reflects group differences concerning the placebo effect by accounting for possible confounding inter-individual differences (e.g. pain sensitivity). This interaction of group and condition became significant (F(2,1172) = 2.72, p=0.007) revealing, as hypothesized, that the HTP group (mean VAS difference of the two conditions 6.95 ± SD 11.31) showed a larger placebo effect than the LTP group (1.16 ± 25.43). These results suggest that levels of prior treatment precision modulate the placebo effect. No main effect of group was observed (p=0.240).

## Integration of expectation and sensory observation

To evaluate the Bayesian framework, we modeled distributions of subjective ratings for every subject separately. Maximum likelihood estimates for the Bayesian model parameters of prior and likelihood were obtained by fitting Gaussian density functions to the data resulting in predictions of the test phase placebo ratings as displayed in *Equation (2)*. By integrating the model parameters of prior and likelihood a prediction of the test phase placebo ratings is presented not only involving the mean but, more importantly, including the respective precision level of prior treatment expectations and sensory inputs reflected in the variance of both normal distributions. As our Bayesian framework incorporates prior and likelihood precisions in a continuous manner, the whole sample (ignoring experimental groups) was analyzed. As expected, the predicted placebo effect

($likelihood\ \mu - posterior\ \mu$) correlated significantly with the observed placebo effect ($mean\ test_{control} - mean\ test_{placebo}$) with r = 0.441, p<0.001. To explain placebo treatment outcomes, the Null model (see Materials and methods) assumed no influence of the treatment experience during conditioning. This was contrasted to the Bayesian integration model hypothesizing the integration of prior expectations and experiences with the new incoming sensory observations (likelihood). By using a random effects (RFX) Bayesian model selection approach (*Rigoux et al., 2014*; *Stephan et al., 2009*) to estimate the overall posterior model probability across subjects, the better explanation of the given data was provided by the Bayesian integration model, reflected in a greater posterior model probability (*Figure 3A and B*, see also *Figure 3—figure supplements 1* and *2* for single subject fits). The RFX conditional expectations of model probabilities of 0.913 (exceedance probability $\varphi_1 \approx$ 100%) for the Bayes model compared to 0.087 (exceedance probability $\varphi_0 \approx$ 0%) for the Null model reflect this result (*Figure 3B*). This finding indicates that the model incorporating the variance of treatment expectation performed better than the Null model.

In more detail, 31 subjects (HTP 17, LTP 14) showed a Bayes factor ($BF_{10}$) larger than three in favor of the Bayesian integration model (*Figure 3A*), indicating at least a moderate evidence (*Kass and Raftery, 1995*; *Lee and Wagenmakers, 2013*). Concerning the Null model, seven subjects showed a Bayes factor ($BF_{01}$) larger than three (HTP 4, LTP 3). This led to a positive evidence ratio of $PER_{10}$ = 31/7 = 4.43. Importantly, no difference between the groups was observed meaning that in both groups, a comparable number of model fits was significantly better described by the Bayesian integration model in contrast to the Null model. In 24 subjects, none of the two models was favored significantly over the other model.

Investigating the mean of treatment expectations ($\mu_{prior}$) and the relationship with placebo effect magnitudes, we did not observe a correlation (p=0.997) which makes it more likely that differences in treatment precision levels are a possible modulator of placebo treatment outcomes. To investigate this further, we used *attraction weight* $w_{prior}$(see Materials and methods, *Equation (3)*), a parameter that considers the precision level (i.e. inverse variance) of both, prior and likelihood, irrespective of the influence of the mean parameters of the two. Importantly, the *attraction weight* reflects a relative, integrated precision measure of prior and likelihood and includes the assumption that a certain level of treatment variability is necessary to induce a placebo effect as full predictability of a treatment outcome would not induce expectation processes (*de la Fuente-Fernández et al., 2004*). In other words, it reflects the relative influence of prior and likelihood on the posterior. Attraction weight was positively correlated with the placebo effect (r = 0.306, p=0.016, *Figure 3C*). This positive relationship indicates that in subjects with higher precision (i.e. less variability) in prior treatment expectation a larger placebo effect magnitude was observed, whereas subjects with higher precision for the perceived sensory inputs (likelihood) showed smaller magnitudes. Using a multiple linear regression approach, our analysis showed that the placebo effect was better predicted by the precision of treatment expectations ($\sigma_{prior}$) compared to the precision of perceived sensory inputs ($\sigma_{like}$) (F(2,59) = 6.83, p=0.002, $R^2$=0.188). The prediction of subjects' placebo effect magnitude was equal to 15.005–0.883*($\sigma_{prior}$)+0.562*($\sigma_{like}$). A negative regression weight for $\sigma_{prior}$ indicates that the placebo effect magnitude is expected to decrease for subjects with less precise (more variable) prior treatment expectations, after controlling for the precision level of sensory inputs ($\sigma_{like}$). Precision of prior treatment expectation was a significant predictor of placebo effect magnitudes (β = −0.883, t(59) = 3.642, p=0.001) and precision of sensory inputs showed a trend effect (β = 0.562, t(59) = 1.842, p=0.070). Only including precision of prior treatment expectation explained less variance ($R^2$=0.141, adj. $R^2$=0.117) than entering both *attraction weight* components ($R^2$=0.188, adj. $R^2$=0.161). This suggests that higher precision or less variability within individual treatment expectations (prior) compared to perceived sensory inputs (likelihood) leads to greater placebo effects.

## Neural correlates of treatment precision

In a next step, we investigated how precision-related treatment outcomes are reflected at the neural level. As we had a strong hypothesis concerning the periaqueductal gray (PAG) (*Roy et al., 2014*; *Yoshida et al., 2013*), we specifically looked for brain activity associated with Bayesian integration parameters within this area. Most importantly, we were interested in the neural correlates of the *attraction weight* ($w_{prior}$) as this value reflects the relative precision of both, expectations and

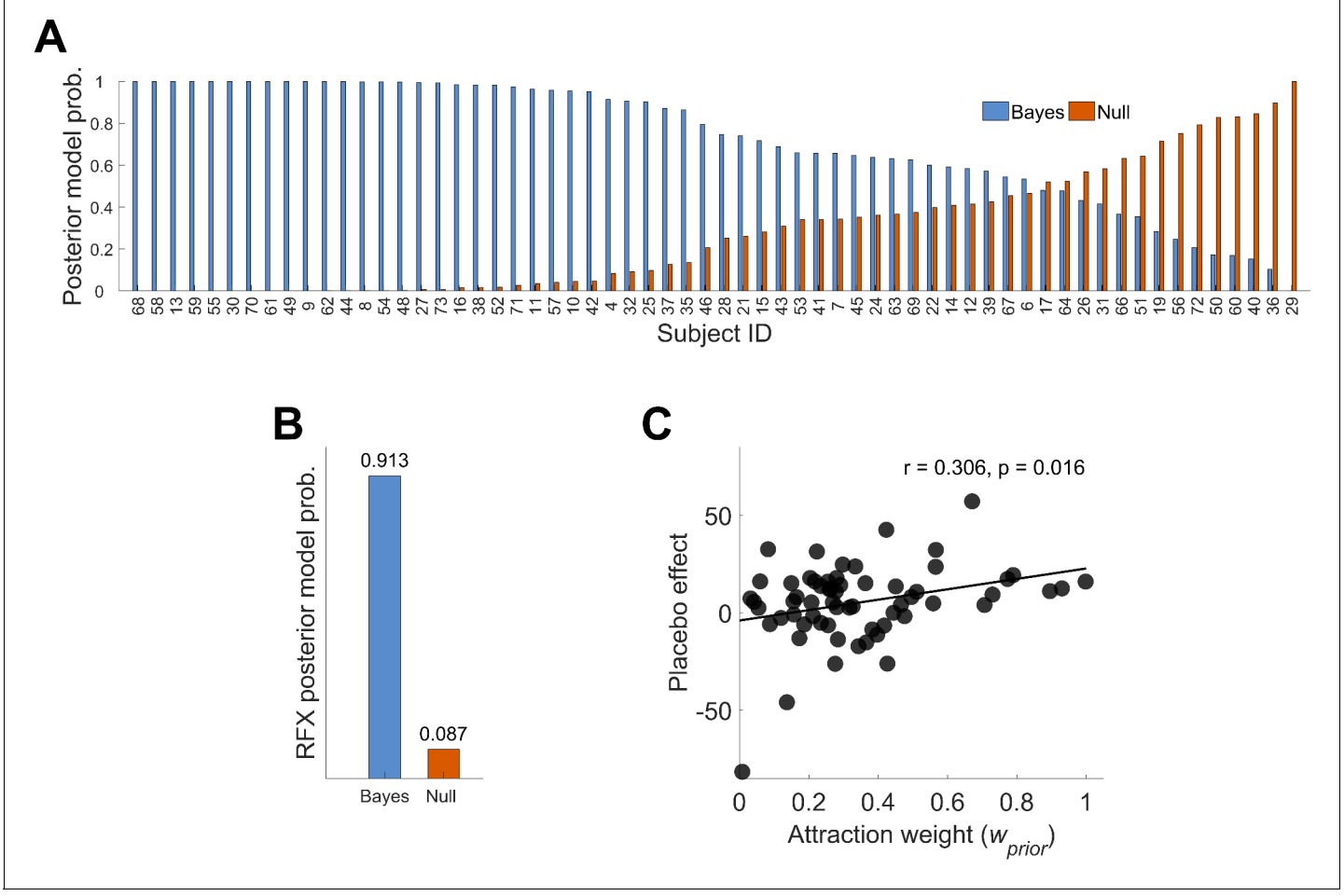

**Figure 3.** Posterior model probabilities given the observed data (N = 62) and relationship between the *attraction weight* and the placebo effect. (**A**) Single subjects posterior model probabilities for the Bayesian and Null model. The data is sorted by the Bayesian model posterior probability of each subject. (**B**) Random effects overall posterior model probability for the two models. (**C**) Positive correlation between the placebo effect and the Bayesian integrated treatment variability (*attraction weight* – relative variability of prior and likelihood, see Materials and methods *Equation (3)*). This is implying that higher treatment precision (prior) compared to higher variability in new sensory inputs (likelihood) may lead to larger placebo effects. See also *Figure 3—figure supplements 1* and *2* for single subject fits. RFX, random effects; r, correlation coefficient.

DOI: https://doi.org/10.7554/eLife.32930.007

The following figure supplements are available for figure 3:

**Figure supplement 1.** Integration of prior and likelihood of single subjects including predictions for the Bayesian and Null model: high treatment precision subjects (HTP group).

DOI: https://doi.org/10.7554/eLife.32930.008

**Figure supplement 2.** Integration of prior and likelihood of single subjects including predictions for the Bayesian and Null model: low treatment precision subjects (LTP group).

DOI: https://doi.org/10.7554/eLife.32930.009

incoming sensory inputs, without any influence of simple intensity coding of pain (no involvement of the mean). By using $w_{prior}$ as a covariate for the test phase placebo condition, we observed an activation in the PAG (coordinates [2 -26 -8], $k_E = 8$, $t(60) = 4.16$, $p_{FWE} = 0.001$; *Figure 4* and *Figure 4—figure supplement 1*). Higher BOLD signals in the PAG were related to smaller *attraction weight* values ($r = -0.457$). This more detailed description of the PAG finding is visualized in *Figure 4B*. In other words, the less precise prior treatment expectations are relative to sensory inputs, the stronger the PAG BOLD signal. Interestingly, the same contrast also revealed a cluster in the rostral ventromedial medulla (RVM) but this was not significant when correcting the p-value for the entire volume (coordinates [2 -36 -46], $k_E = 16$, $t(60) = 4.06$, $p_{uc} < 0.001$). Here, also a negative correlation with the

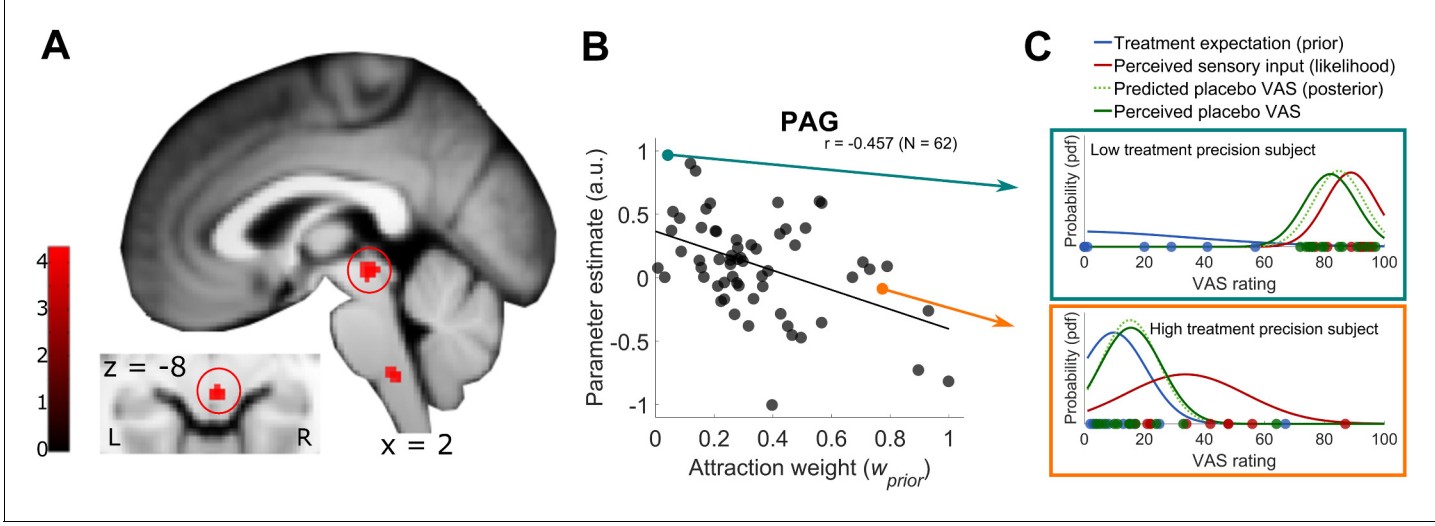

**Figure 4.** Model-based fMRI approach of PAG signal and Bayesian integrated treatment variability. (**A**) One sample t test of the test phase placebo condition with the *attraction weight* as covariate. Beside the PAG activation, a cluster in the rostral ventromedial medulla (RVM) was observed. Visualization was set to $p_{uc}$ <0.001, see also *Figure 4—figure supplement 1* for a more detailed visualization and *Figure 4—source data 1* as well as *Figure 4—source data 2* for first level con-images of the test phase placebo condition. (**B**) Detailed visualization of the negative relationship between PAG BOLD signal and Bayesian *attraction weight* as depicted in A, implying increased PAG signal being associated with higher variability in placebo treatment expectations (prior) and lower variability in new sensory inputs (likelihood). Subject's PAG parameter estimates of the displayed cluster were small volume corrected, extracted per voxel and averaged to result in a mean parameter estimate. (**C**) Single subject examples of the Bayesian integration for a high and low prior treatment subject. Markers (blue – prior, red – likelihood, green – observed data) represent single trial ratings of the respective fitted Gaussian probability density function (pdf). The predicted posterior displays the integrated Bayesian model prediction of the observed data (dashed light green). L, left; R, right; a.u., arbitrary units; PAG, periaqueductal gray; pdf, probability density function; VAS, visual analogue scale.

DOI: https://doi.org/10.7554/eLife.32930.010

The following source data and figure supplements are available for figure 4:

**Source data 1.** First-level con-images of *Figure 4—figure supplement 1* to 4 (HTP group).
DOI: https://doi.org/10.7554/eLife.32930.016

**Source data 2.** First-level con-images of *Figure 4—figure supplement 1* to 4 (LTP group).
DOI: https://doi.org/10.7554/eLife.32930.017

**Figure supplement 1.** Sagittal slices of test phase placebo treatment with the *attraction weight* as covariate (relative precision of prior and likelihood).
DOI: https://doi.org/10.7554/eLife.32930.011

**Figure supplement 2.** Sagittal slices of test phase placebo treatment with the prior μ as covariate (mean of prior treatment expectation).
DOI: https://doi.org/10.7554/eLife.32930.012

**Figure supplement 3.** Sagittal slices of test phase placebo treatment with the log-transformed prior σ as covariate (log-transformed standard deviation of prior treatment expectation).
DOI: https://doi.org/10.7554/eLife.32930.013

**Figure supplement 4.** Exploratory results of test phase placebo treatment with the posterior model probability of Bayes over Null model as covariate (blue bars of *Figure 3A*).
DOI: https://doi.org/10.7554/eLife.32930.014

**Figure supplement 5.** Main pain and placebo effect of the test phase.
DOI: https://doi.org/10.7554/eLife.32930.015

relative precision was observed (r = −0.451). Additional, more explorative, results reflecting the neural correlates of $\mu_{prior}$, $log(\sigma_{prior})$, and the posterior model probability for the Bayesian over the Null model (*Figure 3A*, blue bars) can be found in the Supplement (*Figure 4—figure supplements 2, 3* and *4*).

## Discussion

Our results provide evidence that a Bayesian integration mechanism in the context of placebo hypoalgesia can account for placebo effects on a behavioral level. More importantly, our fMRI data

revealed that key parameters of this mechanism are represented in the periaqueductal gray (PAG). These results add to other approaches also proposing alternative ways of analyzing perceptual experiences such as pain (*Anchisi and Zanon, 2015*; *Büchel et al., 2014*; *Wager et al., 2013*; *Wiech et al., 2014*) indicating the need for models that can parsimoniously account for several modulating factors. Through model-based fMRI, these models allow to identify neural correlates of various aspects of pain perception and placebo hypoalgesia.

On the behavioral level, we observed that placebo effects were smaller in subjects with less precision in their prior treatment experience (higher variability) and more pronounced for those who perceived the treatment as more constant (higher precision) relative to the incoming sensory stimuli. The observed placebo effect did not correlate with the simple mean of prior treatment expectations which reflects the pain intensity during treatment conditioning. This intensity mean of the prior was not able to describe the strength of the placebo effect via a correlative relationship, whereas the precision level of the prior was. By showing that, we were able to present a framework that indicates strong evidence for placebo effects being explained by optimal Bayesian integration (*Anchisi and Zanon, 2015*; *Büchel et al., 2014*).

Also, our results showed that Bayesian integration predicted placebo treatment outcome based on various individual prior and likelihood distributions. The framework did not favor high over low prior precision or vice versa and predicted the different cases equally well. As placebo effect magnitudes are often highly variable (*Vase et al., 2009*) our results reflect the usefulness of a Bayesian approach that can account for variable as well as precise prior treatment expectations. In some subjects, the Bayesian integration approach did not describe the treatment outcome significantly better than the Null model. Reasons for that are highly speculative and include, for example, that not all individuals may combine information of prior experiences and new sensory inputs in an optimal way. Finding markers to predict which individual uses optimal Bayesian integration and which does not would be a subject for future research.

Additionally, the experimental manipulation used in this study provides new insights in the modulation of different levels of precision of prior treatment expectations by holding the test phase stimuli constant in both groups. The present study used test phase stimuli matching prior expectations concerning treatment precision levels in the HTP but not in the LTP group as induced by the different conditioning procedures. This was done to investigate the specific effect of precision levels of prior treatment expectations without interfering variability of the test phase stimulation. However, for that reason, our study cannot answer the question whether a modulation of precision levels of the test phase stimuli may influence the placebo effect and pain perception in a different way. This might be answered in future studies with both matching or mismatching conditioning and test phase precision levels.

Our fMRI data indicated that the PAG signal change represents relative expectation precision of the Bayesian integration process in a placebo hypoalgesia study. This brainstem area is part of the ascending and descending pain system and is crucial for pain modulation (*Fairhurst et al., 2007*), pain avoidance prediction error coding (*Roy et al., 2014*), and the processing of precision of vicarious information (*Yoshida et al., 2013*). The PAG is not only known to mediate pain inhibition (*Jones and Gebhart, 1988*), but also involved in pain facilitatory processes (*Vanegas and Schaible, 2004*) which makes it a key structure of anti- as well as pro-nociceptive effects. Our behavioral data shows that pain modulation underlying placebo hypoalgesia depends on the precision of the prior expectation. Therefore, our data suggests that the PAG is an area crucially involved in precision biased integration processes due to its opposing modulatory properties. Previous research already hinted at a representation of uncertainty in the PAG during a painful vicarious observation task (*Yoshida et al., 2013*). They observed a potent hyperalgesia effect during this vicarious observation task in subjects who showed high susceptibility to induced variability and related this to an increased BOLD signal in the periaqueductal gray (PAG). In this study, the results of the winning uncertainty-hyperalgesia model suggested a strong effect of social assimilation and hyperalgesic uncertainty, but were not able to clearly determine whether this was driven by an underlying linear mean effect of pain or the uncertainty that was induced during the observation task. By using an optimally integrated precision weight to explicitly investigate the influence of variability irrespective of the mean intensity of pain in the placebo treatment context, our data indicates a significant influence of the PAG in variability coding. However, our data does not negate the influence of the mean intensity of

pain as the significant behavioral main effect of condition during our placebo test phase illustrates a clear relationship.

We additionally observed that the rostral ventromedial medulla (RVM) is involved in this process. Again, facilitatory as well as inhibitory projections from the PAG relay in the RVM (*Vanegas and Schaible, 2004*) which underpins and adds to the relationship of variability coding in the PAG as both brainstem areas are involved in opposing modulatory processes during pain. In both, PAG and RVM, a stronger activation was observed when the treatment expectation was less precise (i.e. high variability) relative to the precision of the incoming sensory stimuli. Dependent on the context, including anticipatory and/or direct responses to nociceptive stimuli, the placebo effect literature reports PAG activation patterns being both, increased or decreased in placebo hypoalgesia reflecting the mediating effect of this area (*Eippert et al., 2009*; *Geuter et al., 2013*; *Peciña et al., 2013*; *Scott et al., 2008*; *Wager et al., 2004*; *Wager et al., 2007*; *Wager and Atlas, 2015*; *Zubieta et al., 2005*), whereas the RVM shows mainly increased activation in placebo hypoalgesia (*Eippert et al., 2009*). These activation patterns are likely being related to variability coding as seen in our results. Variability during pain most probably introduces uncertainty about future painfulness (*Seidel et al., 2015*). In this context, our data suggests that less precise prior treatment expectations based on missing information lead to higher activation in the PAG probably due to processing less predictable (*Fairhurst et al., 2007*) outcomes. It seems that the PAG's modulatory processing (*Jones and Gebhart, 1988*; *Linnman et al., 2012*; *Vanegas and Schaible, 2004*) is reflected by this signal increase in the presence of non-precise information about a treatment. Also, some studies were able to show that PAG placebo-induced signal increases were related to the strength of the analgesic effect connecting this to the opioidergic descending pain control system (*Eippert et al., 2009*; *Peciña et al., 2013*; *Wager et al., 2004*). Our results additionally indicate that PAG signal increase is related to less precise prior treatment expectations (prior) compared to more precise incoming sensory stimuli (likelihood) during a placebo treatment, importantly not being related to simple pain intensity but rather optimally integrated variability coding. Therefore, an influence of prior treatment precision levels on opioidergic descending pain modulations is speculated as we also observed the same trend-wise activation pattern in the RVM. This is supported by a previous finding of the anticipated analgesic effect being positively correlated with a signal increase in the PAG (*Scott et al., 2008*), reflecting the responsiveness of this brainstem area to placebo-induced expectations. In both areas, PAG and RVM, placebo-induced BOLD-activations can be significantly reduced by naloxone (*Eippert et al., 2009*), an opioid-antagonist known to impair placebo-dependent pain reduction (*Amanzio and Benedetti, 1999*; *Grevert et al., 1983*; *Levine and Gordon, 1984*; *Levine, 1978*), which reflects their involvement in the opioidergic descending pain control system. As these are the two brain regions we also observed during the coding of expectation precision under a placebo treatment condition, an involvement of the opioidergic descending pain system is likely. Additionally, the PAG-RVM circuit is known to be involved in the aforementioned anti- but also in pro-nociceptive responses such as cholecystokinin (CCK) antagonizing opiate analgesia (*Watkins et al., 1984*), nocebo hyperalgesia (*Benedetti et al., 2006*), opiate hyperalgesia (*Xie et al., 2005*), and safety signal-mediated hyperalgesia (*Wiertelak et al., 1992*). As it was previously speculated that the PAG-RVM modulation in pro-nociceptive responses may likely by involved in the generation and maintenance of discomforting and painful functional disorders such as chronic pain, irritable bowel syndrome, and fibromyalgia (*Tracey and Dunckley, 2004*), a substantial relevance of our findings also connecting the PAG-RVM circuit to treatment precision level coding may shed more light on presumably important aspects of these disorders. A neuronal modulation via the precision level of prior treatment expectation concerning these disorders could generate new hypotheses about inter-individual differences between patients. At this point, these interpretations are highly speculative though and need further investigation in the future.

Due to the study's specific experimental manipulation, the current findings suggest a strong influence of perceptual modulations concerning different levels of prior treatment precision. However, this does not limit the interpretation of the results to only a perceptual but also possible changes in homeostatic regulatory actions as a recent probabilistic model suggested (*Petzschner et al., 2017*; *Stephan et al., 2016*). The authors propose that belief updates are constantly influencing the performance of the interoceptive-allostatic circuit (allostatic self-efficacy) which might be able to better explain differential diagnosis in disorders such as fatigue and depression. This model supports the findings of our study in a way that the precision of prior beliefs predicted placebo treatment

outcomes better than the precision of sensory inputs relating a stronger influence of expectations compared to new sensory information to these disorders.

To summarize, our results add to the existing literature and frameworks, as we observed that not in the cortex but already at the brainstem level, Bayesian integration was able to explain signal changes in placebo hypoalgesia. This nicely compliments previous research suggesting that Bayesian integration during the processing of placebo treatments already takes place at a very basic level of pain perception (*Anchisi and Zanon, 2015*), but, as investigated in a behavioral study, was not able to reveal the underlying neural mechanisms.

In a clinical context, our data clearly indicates that the level of precision of prior treatment experiences and associated expectations is a crucial determinant of placebo effects in treatment outcomes. This mechanism can clinically be exploited by providing precise a priori information concerning a treatment, which will help to create precise prior expectations. Additionally, as there is growing evidence that the chronification of pain may be related to a dysregulation of the descending pain modulatory system (for a review see *Ossipov et al., 2014*), our Bayesian framework and the related neural findings not only add to the placebo literature but also inform other clinical areas investigating disruptions of the modulatory circuits during pain processing.

## Materials and methods

### Participants

Seventy healthy male right-handed subjects with no history of psychiatric or neurological illness were assigned to two groups using a randomized double-blind allocation. Eight subjects had to be excluded due to incomplete data collection or technical difficulties (four in each group). Data analysis was performed on the remaining 62 participants (mean age ± SD: 24.60 ± 3.77 years, range: 19–34 years). Both groups only differed in variability levels within the placebo treatment conditioning (high treatment precision - HTP, no induced variability; low treatment precision - LTP, high induced variability, n = 31 each). The groups did not differ significantly in age (years: HTP 24.71 ± 3.88; LTP 24.48 ± 3.71; t(60) = 0.234, p=0.816) or basic pain thresholds (just painful °C: HTP 44.0 ± 2.4; LTP 44.0 ± 2.3; t(60) = 0.005, p=0.996). They were tested on 2 days with an average break of approximately 5 days between testing (days: HTP 4.55 ± 3.33; LTP 5.03 ± 3.74; t(60) = 0.593, p=0.592). The study was approved by the Ethics Committee of the Medical Board Hamburg, Germany. All subjects were remunerated for participation and gave written informed consent in accordance with the Declaration of Helsinki prior to the experiment. This included information about exclusion criteria (neurological and/or pain related diseases, psychological disorders, skin afflictions, substance abuse, current medication, physical and/or emotional stress) and all experimental procedures such as MRI measurements, thermal and electrical stimulation as well as possible adverse reactions. They were not informed about the purpose of the study investigating placebo hypoalgesia until post-experimental debriefing. Each subject signed a second informed consent after debriefing that the acquired data was not withdrawn from the study.

### Thermal and electrical stimulation

#### Thermal stimulation

To evoke painful sensations, a contact heat stimulator (CHEPS thermode, Medoc, Ramat Yishai, Israel) with a stimulation diameter of 27 mm was used. The thermode was attached to one of four skin patches (*Figure 1C*) of the left volar forearm respectively using a Velcro strap. Each heat stimulus lasted 8 s starting at 32°C baseline, reaching the corresponding target temperature with a rapid heating rate of 70°C/s and returning to baseline by a cooling rate of 40°C/s. After a 2 s delay after stimulus offset, participants rated each stimulus on a visual analogue scale (VAS) with 'no pain' as the left and 'unbearable pain' as the right visual anchor points (corresponding to a VAS of 0 to 100). 'Unbearable pain' was instructed being as painful as subjects would like to remove the thermode from the arm because of too intense heat sensations.

#### Electrical stimulation

As the putative treatment, transcutaneous electrical nerve stimulation (TENS) was introduced. To boost treatment expectations, subjects were told that their individual TENS threshold had to be

detected, as the stimulation in this study needs to be slightly below perceptual threshold. For the actual experimental procedure those thresholds were never used or of any interest. Electrical stimulation was presented using a constant current high-voltage stimulator (DS7A, Digitimer Ltd, Welwyn Garden City, England). By delivering single pulses of 2 ms duration, the individual electrical threshold was acquired on day 2 prior to scanning (all: 0.42 mA; HTP: 0.45 mA; LTP: 0.40 mA). The electrical threshold was defined as a non-painful only just perceptible tickling. During the fMRI experiment each run started with a 2 ms electrical pulse between 0.8 to 0.9 mA and was stated as a resistance measurement of the device to let the subjects experience the putative electrical treatment. After that, the Digitimer was switched off. Post-experimental interviewing affirmed that all subjects sensed the electrical sensation but did not rate it as painful.

## Study design

A between-subjects design was used to create an overall sample that shows sufficient variability of prior precision concerning placebo treatment expectations. All subjects were told that TENS is well-established and known to reduce pain and that the aim of this study was to investigate the underlying neural mechanisms of TENS. To create treatment expectation and let the subjects experience the analgesic effect, both verbal suggestion and conditioning components were used similar to other placebo studies (*Colloca and Benedetti, 2006*; *Montgomery and Kirsch, 1997*; *Price et al., 1999*; *Wager et al., 2004*). Experimental instructions and suggestions concerning the putative treatment were identical for both groups and delivered in written and oral format by an experimenter unaware of the individual's group assignment. The experimenter always presented herself wearing a white coat.

Day 1 was only used to identify potential abnormal pain perception and familiarize the subject with the painful heat stimulation and the VAS rating procedure as well as to check the subject's MR applicability by a physician. First, basic pain thresholds were assessed performing a limits procedure by slowly increasing temperature until the heat was reported as just painful by the subject. This was done three times and the mean was used as an anchor point for the actual calibration procedure trials. Heat calibration consisted of 16 different intensity trials presented in a pseudorandomized order on the right volar forearm. Trials and rating procedure were presented as explained above. To calculate individual temperatures for corresponding VAS ratings of 30, 50, and 70, a sigmoidal function was fitted to the ratings. This ensured that individual pain ratings, despite different intensities, were comparable across participants. The whole procedure during day 1 lasted about 1 hr.

On day 2, the fMRI experiment was performed including a conditioning and test phase. The four skin patches and two TENS electrode positions (*Figure 1C*) were marked on the left volar forearm using a stencil. Each position was used for one of the four experimental sessions (*Figure 1D*) in a pseudorandom order during scanning. Two electrodes for the sham TENS were placed beside the four skin patches. Before going into the scanner, all subjects answered a mood (*Steyer et al., 1997*) and TENS questionnaire to assess their possible foreknowledge regarding TENS as a medical treatment. To account for possible context effects of the MR environment (*Ellerbrock and May, 2015*), the 16-trial-calibration procedure from day one was repeated on the right volar forearm with the subject laying in the scanner without acquiring BOLD data. The new calculated temperatures corresponding to 30 (in °C mean ± SD: HTP 44.4 ± 0.7; LTP 44.3 ± 0.6), 50 (HTP 44.9 ± 0.6; LTP 45.0 ± 0.6), and 70 (HTP 45.5 ± 0.6; LTP 45.7 ± 0.6) VAS were used for scanning which immediately started after pain calibration. For every trial of the whole experiment, all subjects expected a painful heat intensity of 70% of their pain tolerance. During the conditioning phase, all participants underwent two sessions (placebo 'TENS on' skin patch and control 'TENS off' skin patch) with 12 trials each. The order of stimulation (placebo or control session first) and patch position were counterbalanced across participants and matched for the two groups to minimize possible order confounds.

Unbeknownst to the participants, for both groups, the average placebo treatment temperature was set to 30% pain tolerance. Despite that, only within this placebo condition, the two groups differed from each other regarding the level of treatment variability (*Figure 1D*). The HTP group experienced the placebo treatment as consistently effective meaning that they were always presented with the same pain intensity of VAS 30. In contrast, the LTP group received a pain relieving placebo treatment with varying temperatures (SD = 0.55°C; mean VAS 30; range around 30 VAS temperature: ± 0° to ± 0.8°C) across placebo treatment trials. Importantly, during the conditioning, the untreated control stimuli were identical for both groups and included heat stimulation of 70% pain

tolerance intensity without any induced variability across trials. This manipulation procedure served to enhance expectations regarding the placebo treatment and its effectiveness concerning heat pain relief.

Each trial (*Figure 1C*) consisted of an inter-trial-interval (ITI, 12–20 s), followed by an anticipation phase (5.5–8 s), the painful heat stimulation (8 s), a delay (2 s), and the VAS rating procedure (7 s). The ITI was represented by a white fixation cross. For the anticipation cue, either a red fixation cross (control condition) or a red fixation cross surrounded by a yellow circle (placebo condition) was presented. The respective cue remained during heat stimulation and disappeared after cooling down leaving a blank screen for the post-stimulus delay. Subsequently, the VAS appeared and subjects rated and confirmed their perceived painfulness. At the end of each session, participants additionally rated the subjective average painfulness of all trials received during this session.

The conditioning phase was directly followed by the test phase in which the created treatment expectation was compared to the non-manipulated control condition. Importantly, during the test phase, 12 identical heat stimuli of 50% pain tolerance intensity were applied for both placebo and control session, respectively (*Figure 1D*). As stimulation was physically identical in both sessions, placebo effects were assessed by directly comparing VAS ratings of the two conditions. Ratings were expected to be decreased in the placebo compared to the control condition and even more reduced in the HTP compared to the LTP group. Outside the scanner, participants completed several questionnaires to assess personality dimensions and mood components. This also included a post-experimental TENS questionnaire assessing the subject's experience during the treatment prior to debriefing (see *Figure 1—source data 1* and *Figure 1—figure supplement 1*). The whole experimental procedure during day 2 lasted about 3 hr.

## Data acquisition

For stimulus presentation, triggering and, recording of pain ratings, Matlab (Mathworks, Natick, MA) and the open-source Matlab based Psychophysics Toolbox 3 (*Brainard, 1997*; *Pelli, 1997*) was used. Skin conductance was acquired on the distal and proximal hypothenar of the left hand, placing both electrodes on dermatome C8. Additionally during scanning, respiration and heart rate was recorded by using the Expression patient monitoring system (Invivo Corporation, Orlando, FL). A CED 2502 (Cambridge Electronic Design Limited, Cambridge, UK) was used to amplify and a CED micro 1401 to digitalize skin conductance signal at 1000 Hz. The data was recorded by the CED software Spike 2. Magnetic resonance imaging (MRI) data were acquired using a 3T Magnetom Trio scanner (Siemens, Erlangen, Germany) equipped with a 32-channel head coil. BOLD responses were measured using a T2* sensitive echo planar imaging (EPI) sequence. Each volume consisted of 38 transversal slices with a voxel size of $2 \times 2 \times 2$ mm$^3$ and a 1 mm gap (repetition time: 2.35 s, echo time: 26 ms, flip angle: 80°, field of view $224 \times 224$ mm, GRAPPA PAT-factor: 2, reference lines: 48). Volumes were tilted approximately 30° relative to AC-PC line to allow coverage of most of the brainstem area. Considering T1 saturation, the first 4 volumes of every session were discarded. To account for B0 inhomogeneity, prior to each session, B0 field maps were also acquired (40 slices, voxel size: $3 \times 3 \times 3$ mm$^3$, repetition time: 398 ms, short echo time: 4.31 ms, long echo time: 6.77 ms, flip angle: 40°, field of view $216 \times 216$ mm). Additionally, a high-resolution anatomical T1-weighted image was acquired for each subject (MPRAGE sequence, voxel size: $1 \times 1 \times 1$ mm$^3$).

## Data analysis: behavior and modeling

Concerning the statistical analysis, a frequentist approach was applied to test for main and interaction effects of the experiment, and computational modeling was used to predict placebo treatment outcomes and compare them with the observed data.

### Behavior

Data analysis was performed using Matlab (2014a, Mathworks, Natick, MA) and SPSS 24 (IBM, Armonk, NY). A threshold for statistical significance of $p < 0.05$ was used. The placebo effect was calculated by subtracting both means of the two test phase conditions (control - placebo). Individual treatment variability was defined as the subject-specific standard deviation over all 12 trials of placebo ratings during conditioning. The group-specific treatment variability was obtained by averaging over the individuals treatment variabilities. As we expected this to reflect the experimental

manipulation of constant and variable temperatures, it was hypothesized to see higher variability in the ratings of the LTP group compared to lower variability in the HTP group. In general, higher rating variability was assumed to entail low treatment precision and lower rating variability was assumed to reflect higher treatment precision. In a first step, we were interested in experimentally induced treatment variability. Testing this, linear mixed-effects models were performed for pain ratings of conditioning and test phase, respectively. This provided the opportunity to test main effects of condition (placebo vs. control) and group (HTP vs. LTP) as well as the interaction effect by additionally controlling for subject-specific random effects. The analysis was performed on a subset of individuals ($N_{sub}$ = 49). This was done to ensure that the intended manipulation of the respective conditioning procedure of high (HTP) vs. low (LTP) prior treatment precision subjects was induced according to the respective group assignment. To define respective cut-off values (see below), an independent behavioral sample ($N_{ibs}$ = 41; HTP: $n_{ibs-HTP}$ = 21, LTP: $n_{ibs-LTP}$ = 20), that underwent a similar experimental procedure, was used. We selected HTP subjects showing higher rating precision (i.e. smaller variance) than the average LTP group of the independent sample (rating variance/SD < 384.08/19.60). Further, we selected LTP subjects showing lower rating precision (i.e. larger variance) than the average HTP group of the independent sample (rating variance/SD > 194.90/13.96). This led to sub-samples of $n_{sub-HTP}$ = 23 for the HTP and $n_{sub-LTP}$ = 26 for the LTP group with a total of $N_{sub}$ = 49. It is important to note that we used an independent sample to identify cut-off values for subjects in which the experimental manipulation was correctly induced. Furthermore, these cut-off criteria were applied to the conditioning phase data and not to the dependent variable, that is the test phase data. These measures allowed us to avoid any bias towards the initial hypothesis. All other analyses, including the Bayesian model selection and corresponding behavioral as well as fMRI analysis, were performed on the entire sample (N = 62) as optimal Bayesian integration accounts for different precision levels (ignoring experimental groups).

## Computational modeling

For the Bayesian integration, the data was analyzed using a computational model of how individual painfulness of a treatment outcome, $p(pain|sensory\,input)$, is based on prior expectations and experiences, $p(pain)$, as well as new perceived sensory input, $p(sensory\,input|pain)$, and how this may be related to treatment variability (**Figure 1A,B**). In other words, the model predicted the painfulness of the test phase placebo outcome (posterior) by integrating both, the manipulated placebo conditioning ratings (prior) and the test phase control condition ratings (likelihood) using Gaussian probability density functions (pdf):

$$N(\mu_{post}, \sigma_{post}^2) \propto N(\mu_{prior}, \sigma_{prior}^2) * N(\mu_{like}, \sigma_{like}^2) \qquad (2)$$

In a first step, we used an unconstrained non-linear optimization algorithm (implemented in MATLAB's fminsearch function) to fit Gaussian distributions to the rating data. Based on the parameters of the Gaussians, we estimated the posterior according to **Equation (5) and (6)**. We compared this Bayesian integration model to a control model, which only differed in terms of the prior. Assuming no influence of the treatment experience during conditioning, as it is presumed for the Null model, subjects would expect both placebo and control stimuli of the conditioning phase to be on a level of 70% pain tolerance. For that reason, the non-manipulated conditioning control ratings (identical in both groups) were used as a prior for the Null model. This allowed us to compare the model of interest with a Null model by focusing explicitly on the influence of the prior including its precision level (i.e. inverse variance $\frac{1}{\sigma^2}$). To account for individual differences, an *attraction weight* ($w_{prior}$), as displayed in **Equation (3)**, was estimated from the Bayesian integration model reflecting the relative influence of prior ($\frac{1}{\sigma_{prior}^2}$) over likelihood ($\frac{1}{\sigma_{like}^2}$) precision. This weight is larger the more precise (less variable) the prior contribution compared to the likelihood contribution is:

$$w_{prior} = \frac{\frac{1}{\sigma_{prior}^2}}{\frac{1}{\sigma_{prior}^2} + \frac{1}{\sigma_{like}^2}}; w_{prior} = [0, 1] \qquad (3)$$

$$w_{like} = \frac{\frac{1}{\sigma^2_{like}}}{\frac{1}{\sigma^2_{prior}} + \frac{1}{\sigma^2_{like}}} \; ; w_{like} = [0,1] \tag{4}$$

This results in the following equations to estimate Bayesian posterior parameter:

$$\mu_{post} = \mu_{prior} * w_{prior} + \mu_{like} * w_{like} = \frac{\mu_{prior} * \frac{1}{\sigma^2_{prior}} + \mu_{like} * \frac{1}{\sigma^2_{like}}}{\frac{1}{\sigma^2_{prior}} + \frac{1}{\sigma^2_{like}}} \; ; w_{prior} + w_{like} = 1 \tag{5}$$

$$\sigma_{post} = \sqrt{\frac{1}{\frac{1}{\sigma^2_{prior}} + \frac{1}{\sigma^2_{like}}}} \tag{6}$$

Finally, posterior model probabilities and Bayes factors were computed for the two models given the observed data. Bayes factors represent evidence for favoring one model over another reflected in a ratio calculated from the marginal likelihood of each model. Consequently, $BF_{10} = p\left(X|\mu_{Bay}, \sigma^2_{obs}, \sigma^2_{Bay}\right) / p\left(X|\mu_{Null}, \sigma^2_{obs}, \sigma^2_{Null}\right)$, where $BF_{10}$ represents the Bayes factor of the Bayesian integration (1) over the Null model (0). In general, the marginal likelihood is referred to as the model evidence representing the probability of the observed data being produced by the specific model. In the conjugate Gaussian case, the marginal likelihood can be directly estimated (*Demichelis et al., 2006*; *Murphy, 2007*):

$$p\left(X|\mu_{M_x}, \sigma^2_{obs}, \sigma^2_{M_x}\right) = \int \left[\prod_{i=1}^{n} N\left(x_i|\mu_{obs}, \sigma^2_{obs}\right)\right] N\left(\mu_{obs}|\mu_{M_x}, \sigma^2_{M_x}\right) d\mu_{obs} \tag{7}$$

$$= \frac{\sigma_{obs}}{\left(\sqrt{2\pi}\sigma_{obs}\right)^n \sqrt{n\sigma^2_{M_x} + \sigma^2_{obs}}} exp\left(\left(-\frac{\sum_i x_i^2}{2\sigma^2_{obs}} - \frac{\mu^2_{M_x}}{2\sigma^2_{M_x}}\right) + \left(\frac{\frac{\sigma^2_{M_x} n^2 \mu^2_{obs}}{\sigma^2_{obs}} + \frac{\sigma^2_{obs} \mu^2_{M_x}}{\sigma^2_{M_x}} + 2n\mu_{obs}\mu_{M_x}}{2\left(n\sigma^2_{M_x} + \sigma^2_{obs}\right)}\right)\right)$$

$X$ - observed VAS data of placebo test
$\mu_{M_x}/\sigma^2_{M_x}$ - mean/variance of Bayesian integration or Null model
$\sigma^2_{obs}$ - observation noise
$\mu_{obs}$ - observation mean (mean of observed VAS data of placebo test)
$n$ - trials per condition

Assuming that each of the two models was equally likely a priori, the Bayes factor can be directly converted to posterior model probabilities (*Lee and Wagenmakers, 2013*). This leads to a posterior model probability of $p(M_{Bay}|X) = BF_{10}/(BF_{10} + 1)$ for the Bayesian integration over the Null model and a probability of $p(M_{Null}|X) = BF_{01}/(BF_{01} + 1)$ for the Null over the Bayesian integration model. This was performed for each individual subject, respectively. Next, the overall comparison of the two models across subjects was performed. In a first step, both models were compared using the positive evidence ratio (PER). This heuristic serves as an indicator of which model is better at the group level only including subjects that show Bayes factors larger than three for either one of the compared models (*Stephan and Penny, 2007*). PER is the quotient of the number of subjects for which the Bayes factor in favor of the Bayesian integration model ($BF_{10}$) is greater than three, and the respective number of subjects favoring the Null model ($BF_{01}$) with a Bayes factor greater than three. The cut-off value greater three is used as such Bayes factors indicate at least a moderate evidence for one model over another (*Kass and Raftery, 1995*; *Lee and Wagenmakers, 2013*).

In the next step, the single subject log model evidences were used to compute the overall conditional expectations of model probabilities as well as the exceedance probabilities $\varphi_x$ of the model comparison using a random effects (RFX) approach for group studies (*Rigoux et al., 2014*; *Stephan et al., 2009*). This procedure (implemented in SPM's spm_BMS function) assumes that subjects may use different models and thereby allows to control for possible group heterogeneity.

Further, correlations between the model parameters and the placebo effect were estimated by means of the Pearson's product-moment correlation coefficient (r). Additionally, a multiple linear

regression was estimated to describe the relationship between the placebo effect and the *attraction weight* in more detail. The prediction of the placebo effect magnitude was done by inserting the two *attraction weight* components into the regression: the variability of treatment expectation ($\sigma_{prior}$) and sensory input ($\sigma_{like}$).

## Data analysis: fMRI

FMRI data and statistical analyses were performed using statistical parametric mapping (SPM12, Wellcome Trust Centre for Neuroimaging, London, UK). The first four images of each run were discarded prior to further analyses. Preprocessing consisted of motion correction (realignment and field map correction), coregistration of the anatomical T1 image to the functional scans, segmentation of the anatomical T1 image producing DARTEL-imported native tissue class images and in the next step a flow field of the T1 image in Montreal Neurological Institute (MNI) standard space (IXI555_MNI152 template of VBM12 toolbox) using the DARTEL toolbox as implemented in SPM12. First-level analysis was performed in subject-specific native space. Data were high-pass filtered with a 128 s cut-off period and corrected for temporal autocorrelations using a first-order autoregressive model. Functional MRI data analysis was based on a general linear model (GLM) approach as it is implemented in SPM12. The first-level design matrix of each subject consisted of 10 regressors for each session, resulting in a total of forty regressors: anticipation cue onset (5.5–8 s), pain onset (8 s), VAS rating (7 s), six motion regressors obtained during realignment, and one session constant. Each regressor was modeled using a boxcar function and subsequently convolved with the hemodynamic response function. After model estimation, t statistics for each voxel were calculated. All ensuing output images were then normalized to MNI space using the previously obtained subject-specific DARTEL flow field, smoothed with an 6 mm (FWHM) isotropic Gaussian kernel, and then used for second-level analyses. For that, it was investigated whether behavioral Bayesian model parameters reflecting the processing of variability would predict placebo-induced changes in brain signals in the PAG during the test phase placebo condition. Therefore, the *attraction weight* ($w_{prior}$) was used as a covariate in a one-sample t test, testing whether variability variations in prior and likelihood would explain changes in BOLD responses. To complement information about the relationship between Bayesian integration and possible related BOLD responses concerning prior treatment expectation, we additionally performed one-sample t tests either using $\mu_{prior}$, $log(\sigma_{prior})$, or the posterior model probability for the Bayesian over the Null model (as in *Figure 3A*, blue bars) as respective covariates (see *Figure 4—figure supplements 2*, *3* and *4*). *Figure 4—figure supplement 5* shows the main effect of pain and placebo, respectively. For imaging data analyses, results were considered significant after multiple comparison correction using a family-wise error rate (FWE) approach and a threshold of p<0.05. Correction was based on a small-volume approach using a 6 mm sphere around coordinates (MNI, x y z: 1 –29 −12) obtained from previous studies on the PAG (*Linnman et al., 2012*). Other areas found during the analyses were considered significant on a whole brain corrected level of p<0.05. Activations that did not survive whole brain correction but met the criteria of p<0.001 uncorrected and were located in either pain or placebo relevant areas are considered informative and are also reported. Single subject parameter extraction of the BOLD signal and the respective covariate was done by first correcting for multiple comparisons, extracting beta values for each surviving voxel of the corrected ROI cluster per subject, and then averaging subject-wise over all surviving voxels. Statistical maps are presented with a threshold of p<0.001 uncorrected (uc), masked with the field of view of data acquisition, and overlaid on the mean structural image of all subjects. Activations are reported in mm (x y z) using MNI standard space.

## Acknowledgements

The authors would like to express their gratitude to Sophia Thomas for helping with data acquisition and analysis as well as Mareike Menz, Manuel Kuhn, and Jan Mehnert for helpful input concerning data analysis. Additionally, the authors thank Christian Möller for IT and technical support as well as Kathrin Wendt, Katrin Bergholz and Waldemar Schwarz for assisting during MR scanning. We would also like to thank the reviewers Klaas Enno Stephan and Predrag Petrovic for their insightful comments and suggestions during the revision process. This work was supported by the European Research Council, ERC-2010-AdG_20100407 and the German Research Foundation, DFG FOR

1328. Arvina Grahl was supported by the Research Promotion Fund (FFM) for young scientists of the University Medical Center Hamburg-Eppendorf.

## Additional information

### Funding

| Funder | Grant reference number | Author |
|---|---|---|
| University Medical Center Hamburg-Eppendorf | Research Promotion Fund (FFM) for young scientists | Arvina Grahl |
| Deutsche Forschungsge-meinschaft | FOR 1328 | Christian Büchel |
| European Commission | ERC-2010-AdG_20100407 | Christian Büchel |

The funders had no role in study design, data collection and interpretation, or the decision to submit the work for publication.

### Author contributions

Arvina Grahl, Conceptualization, Data curation, Formal analysis, Investigation, Visualization, Methodology, Writing—original draft, Writing—review and editing; Selim Onat, Methodology, Writing—review and editing; Christian Büchel, Conceptualization, Resources, Formal analysis, Supervision, Funding acquisition, Methodology, Writing—review and editing

### Author ORCIDs

Arvina Grahl (iD) http://orcid.org/0000-0002-9699-6329
Selim Onat (iD) http://orcid.org/0000-0002-4782-5603
Christian Büchel (iD) http://orcid.org/0000-0003-1965-906X

### Ethics

Human subjects: The study was approved by and conducted in accordance with the ethics guidelines of the Medical Chamber Hamburg (PV4817). All participants provided informed consent to participate and to publish the data first prior to the experiment and second after debriefing.

### Decision letter and Author response

Decision letter https://doi.org/10.7554/eLife.32930.021
Author response https://doi.org/10.7554/eLife.32930.022

## Additional files

### Supplementary files

• Source data 1. Single trial raw data of VAS ratings including subject ID and group allocation per experimental session (four sessions) for *Figures 2* and *3*.
DOI: https://doi.org/10.7554/eLife.32930.018

• Transparent reporting form
DOI: https://doi.org/10.7554/eLife.32930.019

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
