## [Decision Letter]

Thank you for submitting your article "The periaqueductal gray supports Bayesian integration in placebo analgesia" for consideration by *eLife*. Your article has been favorably evaluated by Timothy Behrens (Senior Editor) and three reviewers, one of whom, Klaas Enno Stephan (Reviewer #1), is a member of our Board of Reviewing Editors. The following individual involved in review of your submission has agreed to reveal his identity: Pedrag Petrovic (Reviewer #3).

The reviewers have discussed the reviews with one another and the Reviewing Editor has drafted this decision to help you prepare a revised submission.

Summary:

This paper combines fMRI with a computational perspective in order to test the hypothesis that the magnitude of placebo effects depend on the precision of treatment expectation. This is a timely and highly important question for placebo research. The study used a clever experimental manipulation, involving heat stimulus delivery under alleged pain reduction by transcutaneous electrical nerve stimulation (TENS), to a large sample (N=62 volunteers). Behavioural analyses indicated evidence in favour of the hypothesis that placebo effects were stronger in individuals with more precise treatment expectations and correlated positively with the precision of prior expectations. In the fMRI analyses, precision of treatment expectation correlated with activity in the periaqueductal gray.

All reviewers found the paper of interest and value. A few conceptual and methodological issues, however, would need to be addressed in a revision. Below, we list our recommendations.

Essential revisions:

1) The paper restricts the effects of belief precision on placebo magnitude to the domain of inference or perception. Inference is certainly a plausible possibility, but it is not the only one. An alternative is that belief precision determines the vigour of regulatory actions (in fact, this would seem more compatible with your fMRI findings, for example, descending anti-nociceptive influences by the PAG). This mechanism, proposed by a recent probabilistic model of homeostatic regulation (Stephan et al., 2016), is consistent with your finding that the precision of treatment expectations predicts the placebo effect better than the precision of sensory inputs (subsection “Integration of expectation and sensory observation”, last paragraph). Second, and related to the previous point, the opening statement: "However, currently there are no unifying perspectives to explain these results" should perhaps be reconsidered. Petzschner et al., 2017 presented a Bayesian framework for describing inference-control loops and – inspired by Büchel et al., 2014 – explicitly hypothesised that precision effects on placebo magnitude could be explained by the mechanism described above (i.e., belief precision determining the vigour of regulatory actions). Their framework also allows for placebo effects to be mediated via belief updates (albeit at levels above homeostatic reflex arcs).

2) It would be helpful if you could provide more details on your Bayesian model selection (BMS) approach. First, how did you compute the marginal likelihood for each subject? Second, could you clarify how you computed the overall Bayes factor of 3.05 – is this simply a product of all the individual Bayes factors? If so, this would correspond to a fixed effects (FFX) analysis (which assumes that all subjects use the same model). This is not wrong per se, but it would be helpful to see a good argument why this should apply in the present situation. A perhaps more appropriate alternative would be to use a random effects (RFX) approach here (for example, the SPM function spm_BMS would allow you to do this). Eyeballing your results in Figure 3, a RFX BMS approach may actually provide stronger evidence in favour of the Bayesian integration model than your current FFX results.

3) The way the equations are presented at the moment will make it difficult for readers unfamiliar with Bayesian methods to recognise the effect of precision. It would be very helpful if you rewrote these equations such that the posterior is represented as a precision-weighted average of prior and likelihood (for example, see the teaching slides by Christoph Mathys for the Zurich Computational Psychiatry Course, e.g. https://www.tnu.ethz.ch/en/teaching/cpcourse/cpcourse2016.html). Similarly, the "attraction weight" will be understood more easily if you rewrite this as a precision ratio; again, see Christoph's slides or Figure 3 by Petzschner et al., 2017.

4) There are two possible reasons related to the study design why a placebo effect was more likely in the HTP group than in the LTP group. First, the testing stimulation resembled the stimulation in the conditioning phase more closely in the HTP than the LTP group. While in the LTP group both stimulations differed with respect to stimulation intensity (VAS 50 vs. VAS 30) and precision (low during conditioning but high during testing), it was only the stimulation that differed in the HTP group (VAS 30 vs. VAS 50) but not the precision, which was high in both phases. In other words, the placebo effects may not only be determined by the (low or high) precision during the conditioning phase but also by the way the effects are assessed in the testing phase. Or in yet other words: Would the results have been different had there been high variability in the testing phase stimulation? The implication for a future experiment would be to use a factorial design, where low vs. high variability would also be used in the testing phase (so that HTP and LTP subjects are equally assigned to conditions that do and do not match the stimulation protocol during the induction phase). We do not expect you to conduct this experiment, of course, but this possible caveat would deserve an explicit discussion in the manuscript.

5) A second reason why a placebo effect was more likely in the HTP group than in the LTP group concerns the relatively long delay between conditioning and testing session of five ± 3 days. This delay emphasized the memory-dependent component of the task, which should produce stronger placebo responses in the condition with the more memorable conditioning experience. If we follow the logic that the higher precision in the HTP group generated a stronger prior, it seems reasonable to assume that participants in this group had a better recollection of their conditioning experience as the LTP group. Even though the memory aspect is inherent to the paradigm as the conditioning always precedes the testing phase, its influence increases the longer the delay is between sessions. In previous studies of the same lab, participants underwent the conditioning and test session either on the same day or on consecutive days. It would be helpful to see a justification of this change in protocol.

6) The group differences in placebo responses are less clear than the Results section is currently suggesting. The placebo effect was defined as the rating difference between the control and the placebo condition during the testing phase. As shown in Figure 2, this difference is indeed larger in the HTP group, but it is driven by a (significant?) group difference in the control rather than the placebo condition. While the stimulation at the placebo site was rated about VAS 47 in both groups, ratings for the control site were higher in the HTP group (about VAS 54) compared to the LTP group (about VAS 48). This suggests a generalization effect or spread of the placebo effect to the control site in the LTP group rather than a stronger placebo effect in the HTP group. As most of the analyses include the differential placebo measure this issue should be addressed in more detail and the interpretation of the results should be carefully adapted.

7) There was some confusion about the exclusion of subjects from parts of the analyses (subsection “Data analysis: Behavior and Modeling”, third paragraph). Could you please clarify in which analyses exactly subjects were excluded and address the potential concern that removing subjects with overly high rating variability in the HTP group or overly low rating variability in the LTP group might bias the result towards the initial hypothesis.

8) The fMRI analysis uses a model-based approach with Bayesian attraction weight as a 2nd level regressor for the placebo analgesia condition. While this is fine, it would seem worthwhile to also use the posterior model probability for the Bayesian (vs. the Null) model (as Figure 3) as regressor in the placebo analgesia condition. This could provide more information on which regions should be studied in order to better understand variability-related placebo response. Furthermore, it would be nice to see some basic contrasts (e.g., main effect of placebo and main effect of pain) as a quality check. This could be presented in the supplementary material.

9) It is not clear whether PAG alone should be the only region of interest in the primary analysis. The authors base their specific focus on PAG on two studies (Yoshida et al., 2013; Roy et al., 2014), and only one of those is more closely related to their model (Yoshida et al., 2013). It would be good to see a more convincing argument why regions such as rostral ACC and lateral OFC are not used as primary ROIs – especially since the priors were partially constructed with higher order abstract information. One could also consider a hierarchical approach in which the primary hypothesis is examined within one region, then within a larger network defined by an independent criterion, and finally an even more explorative search on the full brain volume.

---

## [Author Response]

Essential revisions:1) The paper restricts the effects of belief precision on placebo magnitude to the domain of inference or perception. Inference is certainly a plausible possibility, but it is not the only one. An alternative is that belief precision determines the vigour of regulatory actions (in fact, this would seem more compatible with your fMRI findings, for example, descending anti-nociceptive influences by the PAG). This mechanism, proposed by a recent probabilistic model of homeostatic regulation (Stephan et al., 2016), is consistent with your finding that the precision of treatment expectations predicts the placebo effect better than the precision of sensory inputs (subsection “Integration of expectation and sensory observation”, last paragraph). Second, and related to the previous point, the opening statement: "However, currently there are no unifying perspectives to explain these results" should perhaps be reconsidered. Petzschner et al., 2017 presented a Bayesian framework for describing inference-control loops and – inspired by Büchel et al., 2014 – explicitly hypothesised that precision effects on placebo magnitude could be explained by the mechanism described above (i.e., belief precision determining the vigour of regulatory actions). Their framework also allows for placebo effects to be mediated via belief updates (albeit at levels above homeostatic reflex arcs).

We agree with the reviewers that the interpretation only focusing on the domain of inference or perception is misleading as we certainly do not claim this to be the only possible explanation. However, due to our specific manipulation of the experiment, we focused on this aspect. We added the suggested homeostatic regulation approach to the “Discussion” section (seventh paragraph) to provide a broader view of our results.

Concerning our opening statement (Introduction, first paragraph), we also agree that this might be somehow misleading. We intended to state that different frameworks already use models that parsimoniously try to account for several modulating factors in pain perception but that supporting data is still rare. We changed this statement accordingly.

2) It would be helpful if you could provide more details on your Bayesian model selection (BMS) approach. First, how did you compute the marginal likelihood for each subject? Second, could you clarify how you computed the overall Bayes factor of 3.05 – is this simply a product of all the individual Bayes factors? If so, this would correspond to a fixed effects (FFX) analysis (which assumes that all subjects use the same model). This is not wrong per se, but it would be helpful to see a good argument why this should apply in the present situation. A perhaps more appropriate alternative would be to use a random effects (RFX) approach here (for example, the SPM function spm_BMS would allow you to do this). Eyeballing your results in Figure 3 RFX BMS approach may actually provide stronger evidence in favour of the Bayesian integration model than your current FFX results.

When providing more details with respect to the estimation of the marginal likelihood required for the model comparison, we noticed that the original estimation was incorrect. We have now fixed this and follow the estimation of the marginal likelihood as described in Demichelis et al. (2006) and Murphy (2007). Importantly, this did not change the model comparison results. The neuroimaging results were not affected. Figure 3 as well as Figure 3—figure supplement 1 and 2 have been updated accordingly. We apologize for this error. We now describe the new estimation of the marginal likelihood in detail under “Materials and methods – Data analysis: Behavior and Modeling”. Moreover, we also followed the reviewers’ advice concerning model probabilities according to the RFX model comparison procedure as we agree that this would be the more appropriate approach. Changes, including the corresponding update of Figure 3, can be found in section “Results – Integration of expectation and sensory observation” as well as in the section “Materials and methods – Data analysis: Behavior and Modeling”.

3) The way the equations are presented at the moment will make it difficult for readers unfamiliar with Bayesian methods to recognise the effect of precision. It would be very helpful if you rewrote these equations such that the posterior is represented as a precision-weighted average of prior and likelihood (for example, see the teaching slides by Christoph Mathys for the Zurich Computational Psychiatry Course, e.g. https://www.tnu.ethz.ch/en/teaching/cpcourse/cpcourse2016.html). Similarly, the "attraction weight" will be understood more easily if you rewrite this as a precision ratio; again, see Christoph's slides or Figure 3 by Petzschner et al., 2017.

This might be a misunderstanding, as our initial equation of µpost=µprior*wprior+µlike*wlike represented the posterior as the sum of the precision-weighted average of prior and likelihood (µ representing the average and w the precision weight). Nevertheless, we added different representations of µ_post_ (Equation 5) as well as w_prior_ (Equation 3) and w_like_ (Equation 4) hoping that this satisfies additional concerns.

4) There are two possible reasons related to the study design why a placebo effect was more likely in the HTP group than in the LTP group. First, the testing stimulation resembled the stimulation in the conditioning phase more closely in the HTP than the LTP group. While in the LTP group both stimulations differed with respect to stimulation intensity (VAS 50 vs. VAS 30) and precision (low during conditioning but high during testing), it was only the stimulation that differed in the HTP group (VAS 30 vs. VAS 50) but not the precision, which was high in both phases. In other words, the placebo effects may not only be determined by the (low or high) precision during the conditioning phase but also by the way the effects are assessed in the testing phase. Or in yet other words: Would the results have been different had there been high variability in the testing phase stimulation? The implication for a future experiment would be to use a factorial design, where low vs. high variability would also be used in the testing phase (so that HTP and LTP subjects are equally assigned to conditions that do and do not match the stimulation protocol during the induction phase). We do not expect you to conduct this experiment, of course, but this possible caveat would deserve an explicit discussion in the manuscript.

We agree with the reviewers that our experimental manipulation focusing on investigating levels of precision in prior treatment expectations cannot answer the question concerning the same manipulation in the test phase stimuli. The match vs. mismatch hypothesis is interesting and as suggested by the reviewers is definitely an interesting subsequent study. As suggested, we now discuss this (Discussion, fourth paragraph).

5) A second reason why a placebo effect was more likely in the HTP group than in the LTP group concerns the relatively long delay between conditioning and testing session of five ± 3 days. This delay emphasized the memory-dependent component of the task, which should produce stronger placebo responses in the condition with the more memorable conditioning experience. If we follow the logic that the higher precision in the HTP group generated a stronger prior, it seems reasonable to assume that participants in this group had a better recollection of their conditioning experience as the LTP group. Even though the memory aspect is inherent to the paradigm as the conditioning always precedes the testing phase, its influence increases the longer the delay is between sessions. In previous studies of the same lab, participants underwent the conditioning and test session either on the same day or on consecutive days. It would be helpful to see a justification of this change in protocol.

This seems to be a misunderstanding as our description of the first experimental day was not clear enough. During day 1, participants were only calibrated and familiarized with the heat stimulation. The whole experimental procedure including conditioning and test phase in the MRI scanner was performed on day 2. The test phase directly followed the conditioning phase with a pause of approximately 5 min. We added a more precise wording concerning this in section “Materials and methods – Study design” for clarification.

6) The group differences in placebo responses are less clear than the Results section is currently suggesting. The placebo effect was defined as the rating difference between the control and the placebo condition during the testing phase. As shown in Figure 2, this difference is indeed larger in the HTP group, but it is driven by a (significant?) group difference in the control rather than the placebo condition. While the stimulation at the placebo site was rated about VAS 47 in both groups, ratings for the control site were higher in the HTP group (about VAS 54) compared to the LTP group (about VAS 48). This suggests a generalization effect or spread of the placebo effect to the control site in the LTP group rather than a stronger placebo effect in the HTP group. As most of the analyses include the differential placebo measure this issue should be addressed in more detail and the interpretation of the results should be carefully adapted.

This analysis was performed as we were interested in the effect of the experimentally induced treatment variability as stated in section “Materials and methods – Data analysis: Behavior and Modeling”. Therefore, the focus of this analysis was the interaction reflecting a significant group difference in placebo effects. Especially in a between-subjects design, we cannot define the placebo effect as an absolute mean rating difference in the placebo condition across groups, but only with reference to an individual control condition. Only the interaction term accounts for confounding inter-individual differences such as pain sensitivity etc. The magnitude of pain ratings of one specific condition (control or placebo) is not directly comparable between groups, but the within-subject difference of the two conditions is. We added an addition on this issue in the “Results – Behavioral results” section to address the reviewers concerns.

7) There was some confusion about the exclusion of subjects from parts of the analyses (subsection “Data analysis: Behavior and Modeling”, third paragraph). Could you please clarify in which analyses exactly subjects were excluded and address the potential concern that removing subjects with overly high rating variability in the HTP group or overly low rating variability in the LTP group might bias the result towards the initial hypothesis.

This analysis was performed as we were especially interested in the effect of the experimentally induced treatment variability as stated in section “Materials and methods – Data analysis: Behavior and Modeling”. The intention was to identify effects of highly precise vs. highly variable prior treatment expectations on the placebo response. For this, we had to ensure that our intended manipulation per group was actually correctly induced according to the respective group assignment of the individuals. We used an independent behavioral sample (N_ibs_=41: n_ibs-HTP_=21 and n_ibs-LTP_=20) to create cut-off values as inclusion criteria. Only for the linear mixed-effects models we used the sub-sample (N_sub_=49). All other analysis used the entire sample (N=62) as they all included optimal Bayesian integration which accounts for precision levels (ignoring group assignments). We appreciate the reviewers’ comments concerning the need of addressing this in the manuscript and did so accordingly in the sections “Results – Behavioral results” and “Materials and methods – Data analysis: Behavior and Modeling”.

8) The fMRI analysis uses a model-based approach with Bayesian attraction weight as a 2nd level regressor for the placebo analgesia condition. While this is fine, it would seem worthwhile to also use the posterior model probability for the Bayesian (vs. the Null) model (as Figure 3) as regressor in the placebo analgesia condition. This could provide more information on which regions should be studied in order to better understand variability-related placebo response. Furthermore, it would be nice to see some basic contrasts (e.g., main effect of placebo and main effect of pain) as a quality check. This could be presented in the supplementary material.

Initially, we decided against including the model posterior probabilities as a regressor in our imaging analysis as we thought that this would be quite challenging to interpret. It not only includes the precision levels of prior treatment expectations but also represents all other components of this model comparison such as the means. However, we do agree with the reviewers that for exploratory reasons, such an analysis is useful e.g. to generate new hypotheses to understand the influence of variability on pain perception and placebo hypoalgesia in a Bayesian framework. We now provide the results of this analysis in the “Supplement”. We report whole brain and small volume corrected findings using a correction mask created to address point 9 (see Figure 4—figure supplement 4) to evaluate if the reviewers’ expected rACC and lOFC modulation (“activated due to possible higher order abstract information character of priors”) might appear for the rather complex model comparison parameter compared to the more specific attraction weight (i.e. precision ratio) where we did not observe these activations. Additionally, we provide basic contrasts for the main effect of pain and placebo in the “Supplement” as well.

9) It is not clear whether PAG alone should be the only region of interest in the primary analysis. The authors base their specific focus on PAG on two studies (Yoshida et al., 2013; Roy et al., 2014), and only one of those is more closely related to their model (Yoshida et al., 2013). It would be good to see a more convincing argument why regions such as rostral ACC and lateral OFC are not used as primary ROIs – especially since the priors were partially constructed with higher order abstract information. One could also consider a hierarchical approach in which the primary hypothesis is examined within one region, then within a larger network defined by an independent criterion, and finally an even more explorative search on the full brain volume.

As Yoshida et al. highlighted, the responsiveness to variability coding within the PAG in the context of hyperalgesia represents a very specific mechanism. We believed that Roy et al. were also quite closely related to this as the PAG activation, among others, was representing the influence of expected probability of pain concerning aversive prediction error coding. Combining this with the assumption that prediction error and precision level coding are distinct but share related aspects of modulatory functions, we found the PAG results of Roy et al. quite relevant for our data. To our knowledge, yet, the literature does not show a similar precision level coding mechanism that was first observed by Yoshida et al. in the rostral ACC or lateral OFC. Indeed, it was found via aversive learning tasks, for example by Seymour et al. (2005, Nature Neuroscience), that lateral OFC and rostral ACC signal corresponded to aversive- (not appetitive) specific prediction error modulations during aversive value predictions. However, we believe that these findings focused on very different aspects of pain perception by differentiating appetitive and aversive prediction errors of pain relief which presented more value-related findings rather than investigating the influence of precision levels as our study did. For that reason, our a priori hypothesis only included the PAG as a ROI as this area seemed most promising in the context of precision level coding in a placebo hypoalgesia context. We appreciate the reviewers comment as the manuscript should address these differences more precisely within the “Introduction” section (fifth paragraph). Also, to show the reviewers that our results in the PAG are robust, we created an additional small volume correction mask (see Figure 4—figure supplement 4) not only including the PAG but also the lateral OFC and rostral ACC to correct for multiple comparisons (comparable to a correction procedure where all these regions would be hypothesized a priori as a ROI) resulting in PAG [2 -26 -8], k_e_ = 8, t(60) = 4.16, p_FWE_ = 0.028 compared to p_FWE_ = 0.001 as stated in the manuscript. One small cluster was also found in the left lOFC but did not reach significance threshold (left OFC [-28 38 -20], k_e_ = 2, t(60) = 3.40, p_FWE_ = 0.193).